# An open-source deep learning network AVA-Net for arterial-venous area segmentation in optical coherence tomography angiography

Mansour Abtahi [1], David Le[1], Behrouz Ebrahimi [1], Albert K. Dadzie[1], Jennifer I. Lim[2] & Xincheng Yao [1,2]✉

## Abstract

**Background** Differential artery-vein (AV) analysis in optical coherence tomography angiography (OCTA) holds promise for the early detection of eye diseases. However, currently available methods for AV analysis are limited for binary processing of retinal vasculature in OCTA, without quantitative information of vascular perfusion intensity. This study is to develop and validate a method for quantitative AV analysis of vascular perfusion intensity.

**Method** A deep learning network AVA-Net has been developed for automated AV area (AVA) segmentation in OCTA. Seven new OCTA features, including arterial area (AA), venous area (VA), AVA ratio (AVAR), total perfusion intensity density (T-PID), arterial PID (A-PID), venous PID (V-PID), and arterial-venous PID ratio (AV-PIDR), were extracted and tested for early detection of diabetic retinopathy (DR). Each of these seven features was evaluated for quantitative evaluation of OCTA images from healthy controls, diabetic patients without DR (NoDR), and mild DR.

**Results** It was observed that the area features, i.e., AA, VA and AVAR, can reveal significant differences between the control and mild DR. Vascular perfusion parameters, including T-PID and A-PID, can differentiate mild DR from control group. AV-PIDR can disclose significant differences among all three groups, i.e., control, NoDR, and mild DR. According to Bonferroni correction, the combination of A-PID and AV-PIDR can reveal significant differences in all three groups.

**Conclusions** AVA-Net, which is available on GitHub for open access, enables quantitative AV analysis of AV area and vascular perfusion intensity. Comparative analysis revealed AV-PIDR as the most sensitive feature for OCTA detection of early DR. Ensemble AV feature analysis, e.g., the combination of A-PID and AV-PIDR, can further improve the performance for early DR assessment.

### Plain Language Summary

Some people with diabetes develop diabetic retinopathy, in which the blood flow through the eye changes, resulting in damage to the back of the eye, called the retina. Changes in blood flow can be measured by imaging the eye using a method called optical coherence tomography angiography (OCTA). The authors developed a computer program named AVA-Net that determines changes in blood flow through the eye from OCTA images. The program was tested on images from people with healthy eyes, people with diabetes but no eye disease, and people with mild diabetic retinopathy. Their program found differences between these groups and so could be used to improve diagnosis of people with diabetic retinopathy.

[1] Department of Biomedical Engineering, University of Illinois at Chicago, Chicago, IL 60607, USA. [2] Department of Ophthalmology and Visual Sciences, University of Illinois at Chicago, Chicago, IL 60612, USA. ✉email: xcy@uic.edu

Early disease diagnosis and effective treatment assessment require quantitative analysis of retinal neurovascular changes. Diabetes, strokes, hypertension, and vascular disorders are among the diseases that affect retinal vessels. The blood vessels show abnormalities in the early stages of diabetic retinopathy (DR), including alterations in diameter[1,2]. During the early stages of disease development, as well as throughout the process, arteries and veins are affected differently. Therefore, differential artery-vein (AV) analysis has been shown to be useful for evaluating diabetes, hypertension, strokes, cardiovascular disease, and common retinopathies[3]. The addition of AV analysis capabilities to clinical imaging devices would enhance the accuracy of disease detection and classification. For differential AV analysis, the first step is to perform AV segmentation or classification in retinal images. The AV segmentation has been conducted using a variety of imaging modalities, including fundus photography, OCT, and OCTA[3]. AV segmentation and classification have been mostly performed on color fundus images[3] using feature extraction-based methods[4–6] and machine learning based approaches[7–9]. However, fundus images have limited resolution and contrast to reveal microvascular abnormalities in the retina, particularly difficult for evaluating smaller capillary level blood vessels around the fovea[10,11].

By providing micrometer scale resolution to visualize retinal neurovasculature, OCT has been widely used for ophthalmic research and clinical management of eye conditions[12]. As one new modality of OCT, OCTA detects microvascular distortions associated with eye diseases noninvasively at the capillary level[13]. Especially in the 6 mm × 6 mm and 3 mm × 3 mm field of views, the OCTA can provide depth-resolved visualization of individual retinal layers with the capillary level resolution. Quantitative analysis of the OCTA images has been extensively studied for the objective detection and classification of retinal diseases[14–19]. Through the identification of capillary vortices in the deep capillary plexus, Xu et al.[20] demonstrated how to manually distinguish retinal arteries from veins in clinical OCTA images. Depth resolved OCT profiles were studied by Kim et al.[21] and Adejumo et al.[22] for objective AV classification. Alam et al.[17] and Son et al.[18] analyzed en face OCT and en face OCTA features for AV classification using quantitative feature analysis.

For clinical deployments of differential AV analysis, the development of a fully automated method is essential. Using a convolutional neural network (CNN), Alam et al.[23] demonstrated deep learning AV segmentation with early fusion of en face OCT image and OCTA images for the first time. Using montaged wide-field OCTA images, Gao et al.[24] developed a unimodal strategy in deep learning for AV segmentation. Using different approaches, all above mentioned studies exploring the AV classification or segmentation have been primarily focused on the detection and segmentation of large vessels as arteries or veins. Abtahi et al.[25] quantitatively evaluated multimodal architectures with early and late OCT-OCTA fusions, compared to the unimodal architectures with OCT-only and OCTA-only inputs. They observed that the OCTA-only architecture with OCTA images as input is sufficient for robust AV segmentation. Using 3 mm × 3 mm OCTA images, they were able to segment arteries and veins to the capillary level.

All previous studies for differential AV analysis in OCTA were limited to blood vessel density and caliber quantification. In other words, previously derived AV maps were in binary format to separate arteries and veins, without the preservation of signal intensity, i.e., vascular perfusion information. Recent studies[26–28] indicate that OCTA intensity (non-binarized) derived quantitative features such as flux can provide a better sensitivity to detect vascular perfusion abnormalities, compared to binarized vessel area density (VAD) analysis. In this study, we present a new approach to achieve AV segmentation with preserved OCTA intensity information for AV analysis of the perfusion intensity density (PID). A deep learning network AVA-Net is developed to achieve automatic segmentation of AV areas to generate the AVA map. By multiplying the OCTA image by the AVA map, we can have the OCTA-AV map that contains the OCTA intensity, i.e., vascular perfusion information, with red and blue channels to separate arterial and venous areas. Using this approach, we can classify all the visible vessels in the OCTA images at different orders and scales as arterial or venous. The intersection-over-union (IoU), Dice coefficient, and segmentation accuracy are used as the evaluation metrics for the validation of AVA-Net performance. Seven new quantitative OCTA features, termed arterial area (AA), venous area (VA), AVA ratio (AVAR), total PID (T-PID), arterial PID (A-PID), venous PID (V-PID), and AV PID ratio (AV-PIDR) are verified for objective detection of DR.

## Methods

**Data acquisition.** The study was approved by the Institutional Review Board (IRB) of the University of Illinois at Chicago (UIC) and is in adherence to the ethical standards set forth in the Declaration of Helsinki. The en face OCTA images used for this study are 6 mm × 6 mm scans collected at UIC. In this study, we have two different datasets. The training dataset which consists of 104 OCTA images and their ground truths (68 control, 12 mild DR, 11 moderate DR, and 13 severe DR scans) is planned to be used for training and validation of the CNN. The test dataset which consists of 64 OCTA images without ground truths (25 eyes from 17 control participants, 18 eyes from 13 NoDR patients, and 21 eyes from 18 mild DR patients) is planned to be used for qualitative testing of the CNN and quantitative analysis with the focus on early detection of DR. Table 1 summarizes all participant demographics and diabetes-related parameters in the test dataset. Control subjects and diabetic patients without and with DR in different stages were recruited from the UIC retina clinic. The patients present in this study are representative of a university population of diabetic patients who require clinical diagnosis and management of DR. Subjects who were 18 years of age or older met the inclusion criteria. In addition, diabetic patients having a diagnosis of Type II diabetes mellitus met the inclusion criteria for our diabetic cohort. The diabetic patients were not insulin dependent. Subjects with macular edema, proliferative DR, previous vitrectomy surgery, history of other ocular disorders other than cataracts or minor refractive error, and ungradable and low-quality OCT pictures were exclusion criterions. There is no preference between left or right eyes. A board-certified retina specialist classified the patients as NoDR or different

**Table 1 Demographic of the healthy subjects, NoDR and mild NPDR patients.**

| | Healthy subjects | NoDR | Mild NPDR |
|---|---|---|---|
| Number of subjects (n) | 17 | 13 | 18 |
| Number of images (n) | 25 | 18 | 21 |
| Age (years) | 52.4 ± 14.57 | 56.54 ± 9.21 | 62.28 ± 12.81 |
| Age range | 35–80 | 40–70 | 24–78 |
| Sex (male/female) | 10/7 | 4/9 | 9/9 |
| Duration of diabetes (years) | — | 11.44 ± 5.06 | 18.10 ± 5.50 |
| Diabetes type | — | Type II | Type II |
| Quality index (1–10) | 8.00 ± 0.89 | 8.00 ± 1.11 | 7.43 ± 0.73 |
| HTN prevalence (Y/N) | 0/17 | 11/2 | 15/3 |

*HTN* hypertension.

stages of NPDR according to the Early Treatment Diabetic Retinopathy Study (ETDRS) staging system. All patients underwent a complete anterior and dilated posterior segment examination. All control OCTA images were obtained from healthy volunteers that provided informed consent for OCT/OCTA imaging. Deidentified diabetic datasets were obtained for retrospective analysis. The IRB waived the need for informed consent from the patients, as patient privacy and confidentiality were maintained according to IRB guidelines. All subjects underwent OCT and OCTA imaging of both eyes (OD and OS). One en face OCTA of each eye was used for this study.

Spectral domain (SD) en face OCTA data were acquired using an AngioVue SD-OCT device (Optovue, Fremont, CA, USA). The OCT device had a 70,000 Hz A-scan rate, ~5 μm axial resolution, and ~15 μm lateral resolution for 6 mm × 6 mm scans. Only superficial OCTA images were used in this study. After image reconstruction, en face OCTA was exported from the ReVue software interface (Optovue) for further processing.

**Generating ground truths**. As reported in previous publication[25], readers can rely on various characteristics in OCTA images to manually detect arteries and veins accurately in the 6 mm × 6 mm dataset: (1) The presence of the capillary-free zone is associated with arteries; (2) arteries do not cross other arteries and veins do not cross other veins, physiologically; (3) vessels can be traced back proximally and distally to aid in identification; (4) arteries and veins typically alternate as each vein drains capillary beds perfused by adjacent arteries. Figure 1a, b show a representative OCTA image and corresponding manually generated AV map. For generating AVA maps for the training dataset, the k-nearest neighbor (kNN) classifier is used to classify background pixels in Fig. 1b as pixels in arterial or venous areas. Since we segmented all the visible large vessels in AV maps and used kNN only to classify background pixels as pixels in arterial or venous areas, the generated AVA maps using kNN are reliable and accurate. They were reviewed and approved by an ophthalmologist. Considering Euclidean distance as distance metric and distance-weighted voting, k values between 4 and 25 generate approximately similar and smooth AVA maps. To minimize the computation cost, the k value of 5 is selected. The output of the kNN classifier is presented in Fig. 1c with a lighter tone of blue and red comparing to arteries and veins presented in Fig. 1b. The union of the arteries and veins with corresponding arterial and venous areas generate the AVA maps represented in Fig. 1d. Generating ground truth AV maps for 3 mm × 3 mm OCTA images is discussed in our previous study[25]. The above-mentioned procedure can be used to generate ground truth AVA maps for 3 mm × 3 mm OCTA images.

**Quantitative features**. By multiplying the OCTA image by the AVA map represented in Fig. 1a and d, respectively, we can have the OCTA-AV map demonstrated in Fig. 1e. To the best of our knowledge, for the first time, OCTA-AV maps are introduced in this paper as images that contain the intensity information of an OCTA image, with separate red and blue channels for arterial and venous areas. By using this method, all the vessels at different orders and scales which are visible in the OCTA images can be classified as arteries or veins. Figure 1f shows the fovea (diameter 1 mm) and OCTA layer indicator with a white circle and yellow rectangle, respectively. During image scanning of the macula, the commercial imaging device detects the fovea center automatically to keep the fovea at the center of the image. Accordingly, we can say that the center of the circle is approximately in the center of the image. Since the foveal avascular zone (FAZ) is devoid of blood vessels, the arterial and venous area segmentation in this area is artificial. As with the fovea (diameter 1 mm), the OCTA

layer indicator area at the bottom left corner of the OCTA image is excluded from the OCTA-AV map as well as the AVA map presented in Fig. 1g and h, respectively.

Using the AVA maps and OCTA-AV maps, we can conduct the quantitative analysis for control, NoDR, and mild DR stages. The area of arterial or venous areas can be quantified using the AVA maps. As two novel quantitative features, the percentage of the arterial or venous areas in the total area can be defined as arterial area (AA) or venous area (VA). Therefore, AA and VA can be calculated as follows

$$\alpha_A = 100 \times \frac{A_A}{A_T} \tag{1}$$

$$\alpha_V = 100 \times \frac{A_V}{A_T} \tag{2}$$

where $A_A$, $A_V$, and $A_T$ are arterial, venous, and total area in AVA maps, respectively, and $\alpha_A$ and $\alpha_V$ are AA and VA, respectively. To calculate AA or VA, the number of pixels in the arterial or venous area can be divided by the number of total pixels multiplied by 100. Since the summation of arterial and venous areas are the total area, mathematically we have the following relationship between AA and VA

$$\alpha_A = 100 - \alpha_V \tag{3}$$

We also can define the arterial-venous area ratio (AVAR), $\alpha_{AV}$, as follows

$$\alpha_{AV} = \frac{A_A}{A_V} = \frac{\alpha_A}{\alpha_V} \tag{4}$$

Most commonly, the binarized OCTA images are used to calculate VAD, also known as vessel density (VD), perfusion density (PD), blood vessel density (BVD), and capillary density[13,29]. VAD in binarized OCTA images is the ratio of the area occupied by vessels divided by the total area converted to a percentage. In this paper, using the OCTA-AV maps that contain the OCTA intensity information, we define perfusion intensity density (PID) as a novel quantitative feature that does not require binarization with any thresholding method. The mean of the pixel intensities converted to a percentage in the total area, arterial area, and venous area can be defined as total PID (T-PID), arterial PID (A-PID), and venous PID (V-PID), respectively. So, T-PID, A-PID, and V-PID can be calculated as follows

$$P_T = \frac{100}{255} \times \frac{\text{summation of intensities in the total area}}{\text{total number of pixels}} = \frac{100}{255} \times \frac{1}{A_T} \sum_{A_T} I \tag{5}$$

$$P_A = \frac{100}{255} \times \frac{\text{summation of intensities in arterial areas}}{\text{total number of pixels in arterial areas}} = \frac{100}{255} \times \frac{1}{A_A} \sum_{A_A} I \tag{6}$$

$$P_V = \frac{100}{255} \times \frac{\text{summation of intensities in venous areas}}{\text{total number of pixels in venous areas}} = \frac{100}{255} \times \frac{1}{A_V} \sum_{A_V} I \tag{7}$$

where $P_A$, $P_V$, and $P_T$ are A-PID, V-PID, and T-PID, respectively, and $I$ is intensity values in OCTA-AV maps. T-PID represents quantitative feature calculation without differentiation of the AV areas, while A-PID and V-PID represent quantitative features after AV area segmentation for differential AV analysis. The arterial-venous PID ratio (AV-PIDR), $P_{AV}$, can also be defined as the division of the A-PID by V-PID as formulated bellow

$$P_{AV} = \frac{P_A}{P_V} \tag{8}$$

To the best of our knowledge, we defined seven new quantitative OCTA features (AA, VA, AVAR, T-PID, A-PID, V-PID, and AV-PIDR) related to AVA maps and OCTA-AV maps that can be used for quantitative analysis of diseases at different stages. The processes for calculating these quantitative features are shown in Fig. 1i–n, with blue arrows and boxes. We

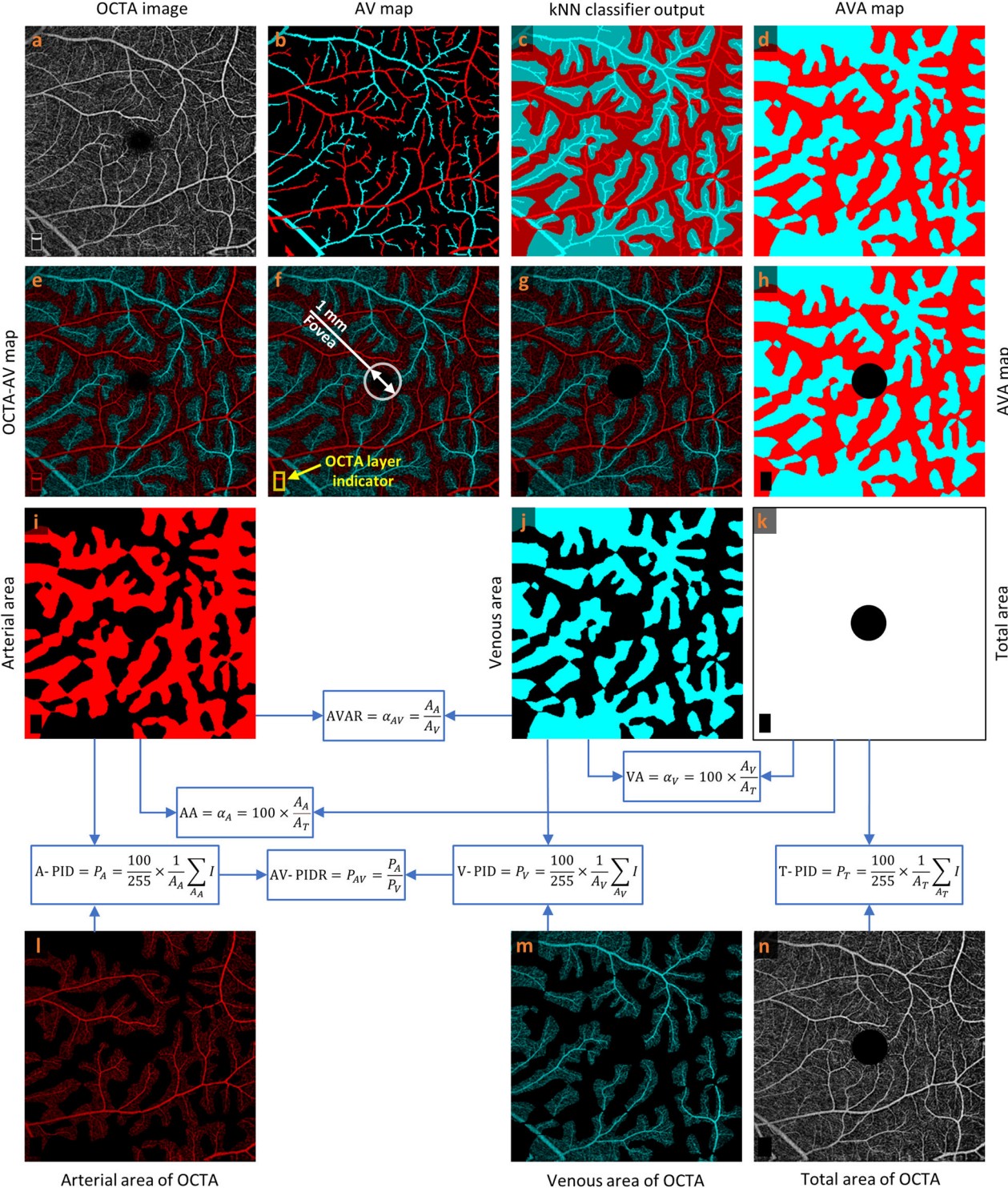

**Fig. 1 Illustration of basic procedures for generating AVA map from OCTA image and calculating quantitative features. a** OCTA image. **b** Manually generated AV map. **c** Output of kNN classifier. **d** AVA map. **e** OCTA-AV map. **f** Fovea and OCTA layer indicator in the OCTA-AV map. **g** OCTA-AV map excluding the fovea and OCTA layer indicator area. **h** AVA map excluding the fovea and OCTA layer indicator area. **i** Arterial area. **j** Venous area. **k** Total area shows the summation of arterial and venous areas with white color. **l** Arterial area of the OCTA image. **m** Venous area of the OCTA image. **n** Total area of the OCTA image excluding the fovea and OCTA layer indicator area. Calculating quantitative features is indicated by blue arrows and boxes. OCTA optical coherence tomography angiography, AV artery-vein, kNN k-nearest neighbor, AVA arterial-venous area, AVAR $\alpha_{AV}$, arterial-venous area ratio, AA $\alpha_A$, arterial area, VA $\alpha_V$, venous area, A-PID $P_A$, arterial perfusion intensity density, V-PID $P_V$, venous perfusion intensity density, T-PID $P_T$, total perfusion intensity density, AV-PIDR $P_{AV}$, arterial-venous perfusion intensity density ratio.

are going to report these quantitative features in the whole image for control, NoDR, and mild DR subjects to show their importance. These quantitative features can also be measured and reported in other regions of the OCTA images such as parafovea and perifovea, as well as their quadrants, however, that is beyond the scope of this paper.

**Statistical analyses**. For the statistical analysis of quantitative features, due to the limited dataset size, we treated each eye as a single observation for some subjects with images of both eyes. We performed the Shapiro–Wilk test to check if quantitative features are normally distributed. We used $\chi^2$ tests to assess the distribution of sex and hypertension among different groups. Age and quality index distributions were compared using analysis of variance (ANOVA). One-versus-one comparison of quantitative features between different groups was performed by the unpaired two-sided Student's $t$ test. We also applied the Bonferroni correction to compare the difference in mean values of the quantitative features. $P < 0.05$ was considered statistically significant.

**AVA-Net architecture**. For fully automated AVA segmentation in en face OCTA images, we propose the AVA-Net, a U-Net-like architecture, illustrated in Fig. 2. The input of the AVA-Net is the grayscale OCTA image. The OCTA image contains the information of blood flow strength and vessel geometry features. Since there are two classes for segmentation: arterial areas and venous areas, this is a binary segmentation problem. So, the output of AVA-Net is a single-channel grayscale image in which arterial pixels are denoted by 1 and venous pixels are denoted by 0. In this article, they are shown in blue and red colors for better demonstration. The overall design of the AVA-Net is composed of an encoder to extract features from the image and a decoder to construct the AVA map from the encoded features. The encoder includes 5 encoder blocks to reduce the image resolution by downsampling. As shown in Fig. 2b, encoder blocks composed of two $3 \times 3$ convolution operations, 4 dilated convolution operations in parallel with dilation rates from 2 to 5, a concatenation operation, and a max-pooling operation with a pooling size of $2 \times 2$.

On the other hand, the decoder is composed of 5 decoder blocks, followed by 2 CBR (Conv - Batch Normalization - ReLU) blocks, and final convolutional operation with a sigmoid activation function. As shown in Fig. 2b, CBR block is composed of a $3 \times 3$ convolution operation, followed by a batch normalization and a ReLU activation function. The decoder blocks are composed of 2 CBR blocks, up-sampling, and concatenation operation that concatenate generated features with features coming from encoders blocks by skip connections. In all the five levels, the features prior to the max-pooling operation in the encoder blocks is transferred to the decoder blocks by skip connections. These feature maps are then concatenated with the output of the up-sampling operation in the decoder block, and the concatenated feature map is propagated to the successive layers. These skip connections allow the network to retrieve the information lost by max-pooling operations. Details of the different operations in the AVA-Net layers are presented in Fig. 2.

**Loss function**. In this study, the CNN was trained using IoU loss[30] or Jaccard loss to directly optimize the IoU score, the most commonly used evaluation metric in segmentation[31]. For multi-class segmentation, it is defined by

$$L_{IoU} = 1 - \frac{\sum_{c=1}^{C}\sum_{i=1}^{N}g_i^c s_i^c}{\sum_{c=1}^{C}\sum_{i=1}^{N}(g_i^c + s_i^c - g_i^c s_i^c)} \quad (9)$$

where $g_i^c$ is the ground truth binary indicator of class label $c$ of voxel $i$, and $s_i^c$ is the corresponding predicted segmentation

probability. $N$ is the number of voxels in the image and $C$ is the number of classes. Since we have two classes, this is a binary segmentation problem. So, we have

$$L_{IoU} = 1 - \frac{\sum_{i=1}^{N}g_i s_i}{\sum_{i=1}^{N}(g_i + s_i - g_i s_i)} \quad (10)$$

**Training process**. AVA-Net architecture was trained using the Adam optimizer with a learning rate of 0.0001 ($\beta_1 = 0.9$, $\beta_2 = 0.999$, $\epsilon = 10^{-7}$) to have a smooth learning curve for the validation dataset. With the IoU loss function, mini-batch sizes of 28 were utilized for the training. The training process takes up to 5000 epochs when the model performance stops improving on the validation dataset. One epoch is defined as the iteration over 3 training batches. To avoid overfitting, data augmentation methods are applied on the fly during training, including random flipping along horizontal and vertical axes, random zooming, random rotation, random image shifting, random shearing, random brightness shifting. As retinal vessels in OCTA images have diverse tree-like patterns and differing vessel diameters, and because images can be taken from different locations of right and left eyes with diverse quality, the above-mentioned data augmentation methods could help improve the generalization capability of the CNN for segmenting unseen images. Since our data is limited, the 5-fold cross-validation procedure is used for CNN evaluation after shuffling the data based on the patient's eye. Therefore, in each fold, the network was trained with 80 percent of the images, and evaluation was performed on the other 20 percent of the images.

The IoU and Dice coefficient metrics, also known as the Jaccard Index and F1 Score, respectively, are mostly used in image segmentation. Therefore, in addition to segmentation accuracy, IoU and Dice coefficient were used as the evaluation metrics for evaluating the AVA segmentation, by comparing the predicted AVA maps with manually labeled ground truths.

The training was performed on a Windows 10 computer using NVIDIA Quadro RTX 6000 Graphics Processing Units (GPU). The CNN was trained and evaluated on Python (v3.9.6) using Keras (2.6.0) with Tensorflow (v2.6.0) backend. Training every fold of AVA-Net took about 10 h with the mentioned resources.

**Reporting summary**. Further information on research design is available in the Nature Portfolio Reporting Summary linked to this article.

## Results

**AVA-Net performance**. We performed 5-fold cross-validation on the training dataset, using 80 percent of the images for training and 20 percent of them for validating AVA-Net. Table 2 presents the average and standard deviation of the IoU and Dice, as well as accuracy for AVA-Net. In Fig. 3, the visual results of the AVA segmentation achieved by AVA-Net for six (three control, one mild, one moderate, and one severe samples) validation samples in the training dataset are illustrated. Figure 3 presents OCTA images, ground truth of AVA maps, predicted AVA maps, the ground truth of OCTA-AV maps (GT-OCTA-AV maps), and predicted OCTA-AV maps in the rows from top to bottom, respectively. It can be concluded that the predicted AVA maps for healthy and NPDR subjects are highly consistent with the ground truth. That means AVA-Net is able to detect and classify arteries and veins, and their corresponding areas. However, there are some incorrect segmentations that are shown by yellow rectangles in Fig. 3. The visual performance of AVA-Net for segmenting representative OCTA images in the test dataset is presented in Fig. 4.

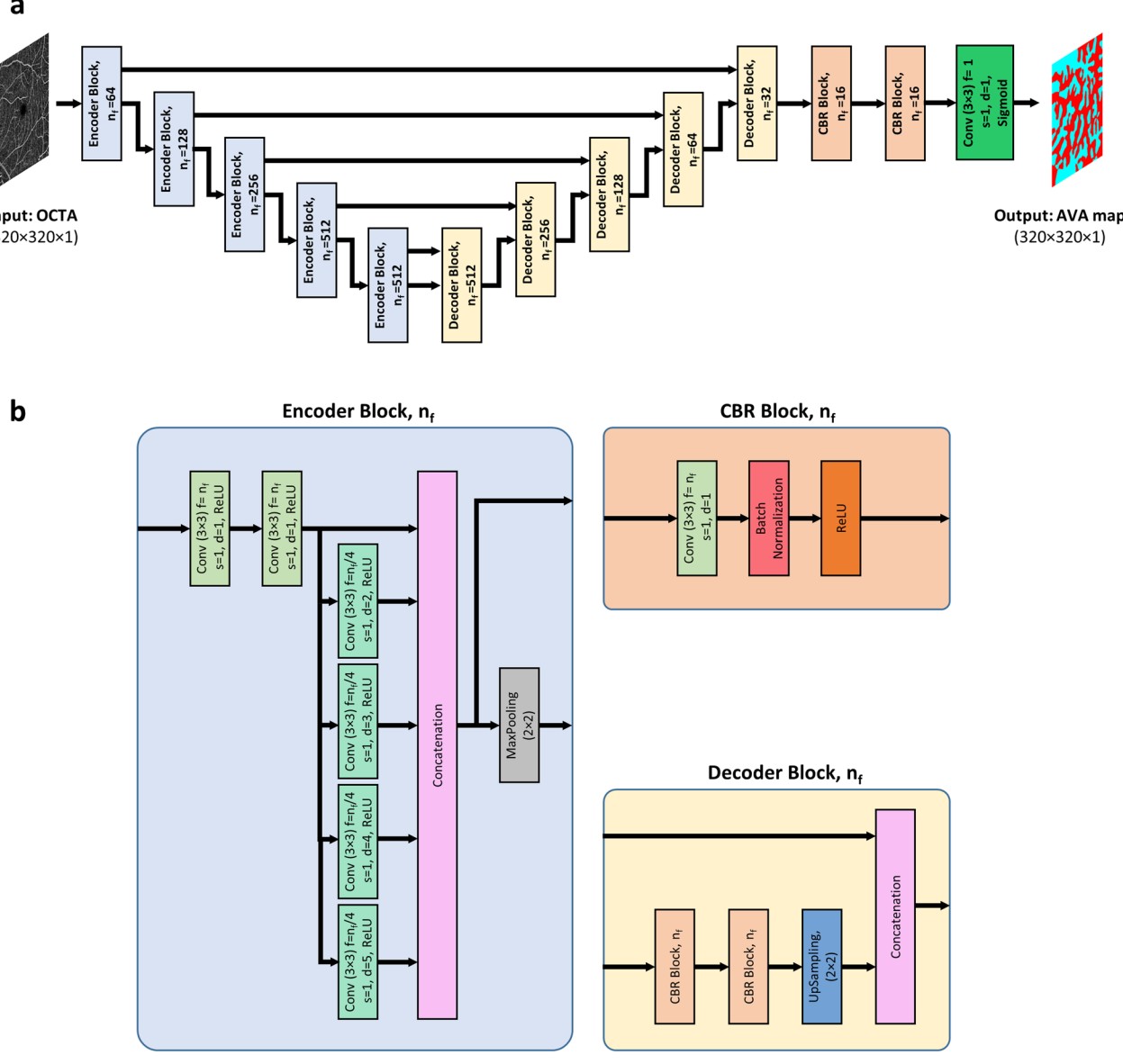

**Fig. 2 AVA-Net architecture. a** overview of the blocks in AVA-Net architecture. **b** the individual blocks that comprises AVA-Net. In this figure, Conv, f, s, d, and $n_f$ stand for convolution operation, number of filters in the convolution, strides of the convolution, dilation rate of the convolution, and number of filters specified for the corresponding block, respectively. OCTA optical coherence tomography angiography, AVA arterial-venous area, CBR Conv - Batch Normalization – ReLU.

| Table 2 Performance results of AVA-Net. | | |
|---|---|---|
| **Mean (±std dev)** | | |
| IoU (%) | Dice (%) | Accuracy (%) |
| 78.02 (±0.54) | 87.65 (±0.34) | 86.33 (±0.20) |

**Quantitative analysis**. Studying the test dataset with demographic details presented in Table 1, no statistically significant differences were found between the different groups with regards to age, quality index, and sex (ANOVA, $p = 0.077$, ANOVA, $p = 0.079$, and $\chi^2$ test, $p = 0.305$). Furthermore, there was no difference in hypertension across diabetes groups ($\chi^2$ test, $p = 0.92$). Between the diabetic groups, there was a significant difference in the duration of diabetes (Student's $t$ test, $p = 0.019$). The diabetic patients in this study were not insulin dependent.

The AA, VA, AVAR, T-PID, A-PID, V-PID, and AV-PIDR in the whole image for the test dataset are calculated as described in section 2.3 using the AVA maps predicted by the AVA-Net and OCTA-AV maps produced after that. Shapiro–Wilk test indicated that all features were normally distributed. Thus, we performed individual pairwise comparisons using an unpaired two-sided Student's $t$ test. Significant differences between groups corresponding to $P < 0.05$, $P < 0.01$, and $P < 0.001$ are denoted by *, **, and ***, respectively. To reduce the occurrence of false positives in multiple hypothesis testing, we applied a Bonferroni correction as a conservative method for probability thresholding. Applying the Bonferroni correction, the statistical significance of the P value between the three groups was adjusted as $P < 0.0167$, and significant differences are denoted by the † symbol. Figure 5 presents the boxplots of the AA, VA, and AVAR for control, NoDR, and mild DR subjects in the whole image. Based on Eq. (3), the summation of AA and VA is 100. Thus, an increase in AA means a decrease in VA with the same value. Thus, the P values related to AA and VA features are identical. We observed a 3.6% increase in whole image AA for the mild DR group compared to

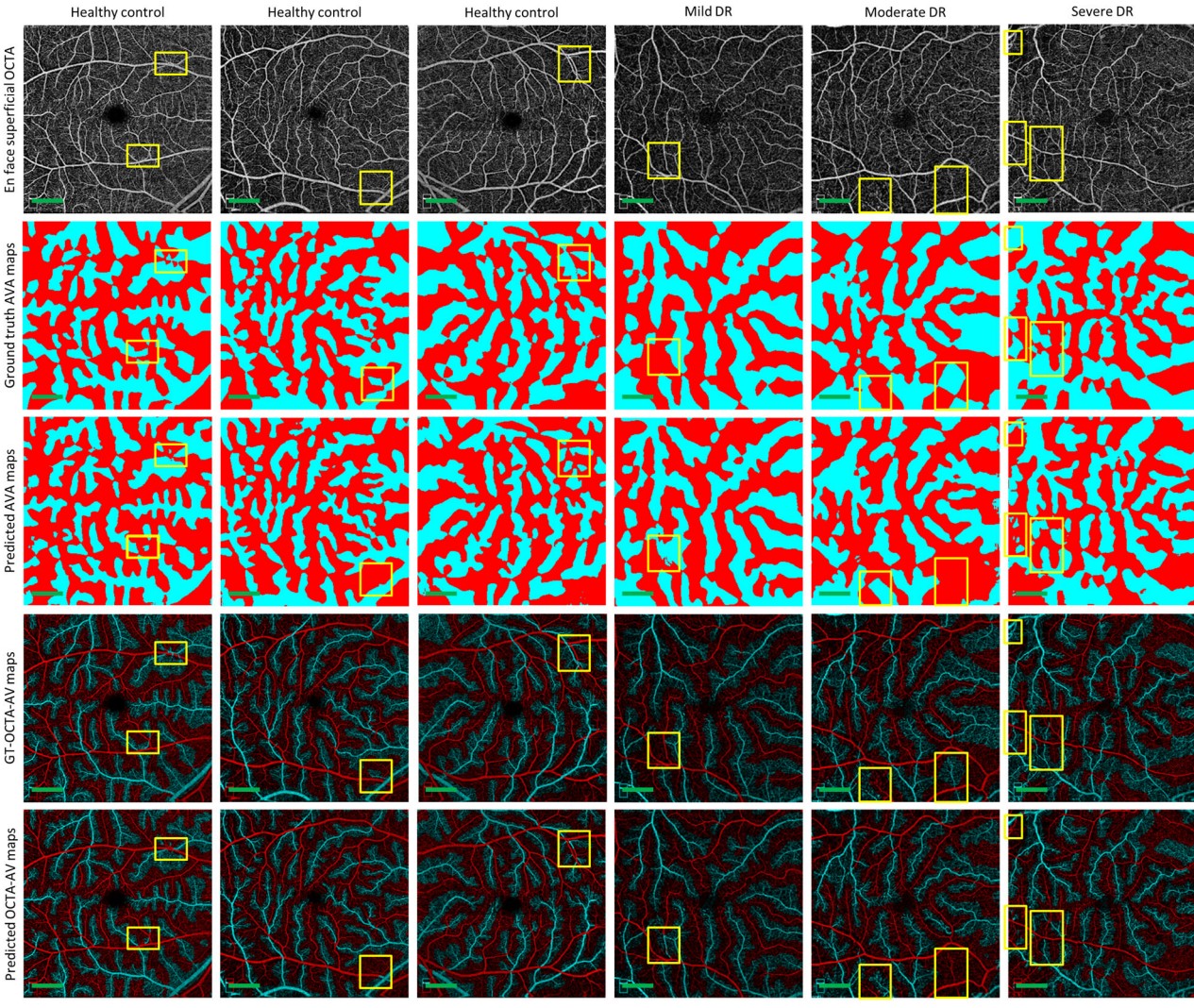

**Fig. 3 Comparative illustration of the AVA segmentation performance achieved by AVA-Net trained on the training dataset.** Each column shows a different sample. Row 1 OCTA images. Row 2 Ground truth AVA maps. Row 3 Predicted AVA maps by AVA-Net. Row 4 GT-OCTA-AV maps generated by multiplying the OCTA images by ground truth AVA maps. Row 5 predicted OCTA-AV maps generated by multiplying the OCTA images by predicted AVA maps. Yellow rectangles indicate some areas that are segmented incorrectly. DR diabetic retinopathy, OCTA optical coherence tomography angiography, AVA arterial-venous area, AV artery-vein, GT ground truth; Scale bar is 1 mm.

the control group. While VA decreased by 4.0% for the mild DR group compared to the control group. The AVAR analysis in Fig. 5b further confirms the observation with an increase of 7.6% for the mild DR group. The Bonferroni correction indicates that none of these above-mentioned changes are significant.

The boxplots of the T-PID, A-PID, V-PID, and AV-PIDR for control, NoDR, and mild DR groups in the whole image are presented in Fig. 6. T-PID and A-PID respectively showed 6.1 and 7.2% decreases from control to mild DR groups but no significant differences between control and NoDR eyes. We observed a 7.4% decrease in V-PID for mild DR groups compared to the NoDR group. Differential AV PID analysis reveals that the A-PID decreases, but V-PID increases in NoDR subjects compared to control subjects. Because of the opposite AV changes, the relative AV-PIDR shown in Fig. 6b provides a sensitive metric to differentiate control, NoDR, and mild DR groups from each other. We observed 5.1 and 2.6% decreases in the whole image AV-PIDR, respectively, for NoDR and mild DR groups compared to the control group, and a 2.6% increase from NoDR to mild DR groups. Based on the Bonferroni correction, changes in A-PID from control to mild DR and changes in AV-PIDR from control to mild DR and

from NoDR to mild DR are significant. According to Bonferroni correction, A-PID and AV-PIDR together can show significant changes in all three pairs.

## Discussion

Differential AV analysis has been demonstrated to be important for evaluating diabetes, hypertension, strokes, cardiovascular disease, and common retinopathies[3]. The addition of AV analysis capabilities to clinical imaging devices would enhance the accuracy of disease detection and classification. The segmentation of AV has been conducted using a variety of imaging modalities, including fundus photography, OCT, and OCTA. Traditional OCT and fundus images are limited in their ability to detect microvascular abnormalities at capillary level. As a new OCT modality, OCTA provides a noninvasive method of detecting microvascular distortions associated with eye diseases with capillary level resolution. Using feature extraction-based methods and machine learning based approaches, all previous studies exploring the AV classification or segmentation have been primarily focused on the detection and segmentation of large vessels as arteries or veins.

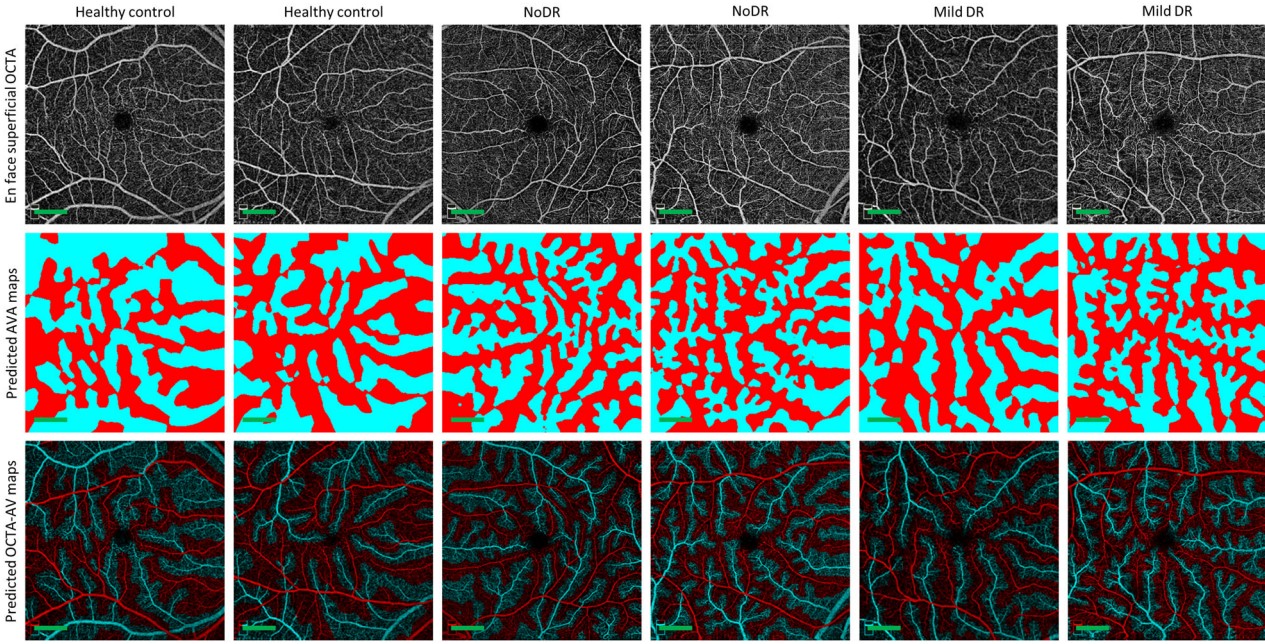

**Fig. 4 The performance of the AVA-Net on representative samples in the test dataset.** Each column shows a different sample. Row 1 OCTA images. Row 2 Predicted AVA maps by AVA-Net. Row 3 predicted OCTA-AV maps. DR diabetic retinopathy, NoDR diabetic patients without DR, OCTA optical coherence tomography angiography, AVA arterial-venous area; Scale bar is 1 mm.

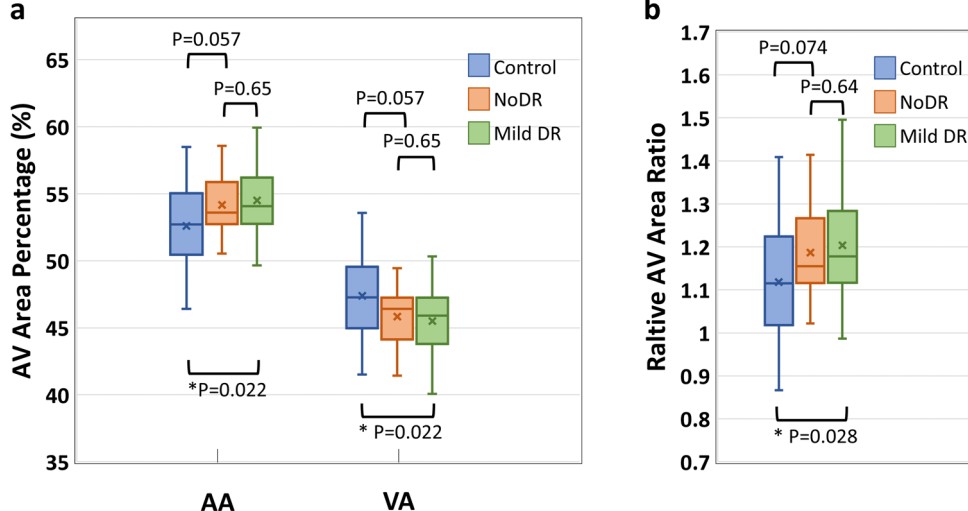

**Fig. 5 Boxplot of area related quantitative features for control, NoDR, and mild DR groups. a** boxplot of AA and VA. **b** boxplot of AVAR. *Significant at $P < 0.05$. Each box indicates the interquartile range (top: the third quartile; bottom: the first quartile) with the whiskers extending to the most extreme points and with a horizontal line and cross mark indicating the median and mean, respectively. The number of samples used for the analysis is $N_{Control} = 25$, $N_{NoDR} = 18$ and $N_{mild} = 21$. AV artery-vein, AA arterial area, VA venous area, DR diabetic retinopathy, NoDR diabetic patients without DR.

In this study, we employed a deep learning network AVA-Net for AVA segmentation in OCTA. By multiplying the OCTA image by the AVA map generated by AVA-Net, we have the OCTA-AV map that contains highly detailed vasculature maps of the OCTA image, with separate red and blue channels for arterial and venous areas. In other words, using this method, we can segment all the vessels in the OCTA image up to the capillary level as arterial and venous. By using the OCTA-AV map, all OCTA-related features previously reported in the literature such as VAD, vessel length density (VLD), vessel diameter index (VDI), vessel tortuosity (VT), and branchpoint density (BD) can be calculated separately for arterial and venous areas to check the effectiveness of the AV analysis.

For fully automated AVA segmentation using OCTA images, we have developed the AVA-Net, a U-Net-like architecture. U-Net-like architectures are commonly used for biomedical image segmentation because they produce reliable and highly accurate results with small datasets. In AVA-Net, we employed encoder blocks containing dilated convolutional operations connected to decoder blocks. We used accuracy, Dice score, and IoU metrics to assess the AVA-Net performance. The results of the cross-validation study revealed the AVA-Net performed well in AVA segmentation by achieving an accuracy of 86.33% and a mean IOU of 78.02%, and a mean Dice score of 87.65% on the validation dataset. Qualitatively AVA-Net has a good AVA segmentation performance on the validation and test dataset.

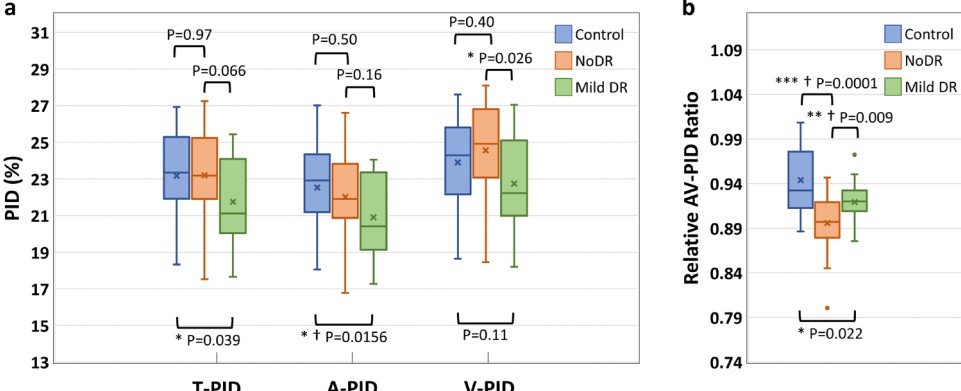

**Fig. 6 Boxplot of perfusion intensity related quantitative features for control, NoDR, and mild DR groups. a** Boxplot of T-PID, A-PID, and V-PID. **b** Boxplot of AV-PIDR. *Significant at $P < 0.05$. **Significant at $P < 0.01$. ***Significant at $P < 0.001$. † Bonferroni correction significant at $P < 0.0167$. Each box indicates the interquartile range (top: the third quartile; bottom: the first quartile) with the whiskers extending to the most extreme points and with a horizontal line and cross mark indicating the median and mean, respectively. The number of samples used for the analysis is $N_{Control} = 25$, $N_{NoDR} = 18$ and $N_{mild} = 21$. PID perfusion intensity density, T-PID total PID, A-PID arterial PID, V-PID venous PID, AV-PID arterial-venous PID, DR diabetic retinopathy, NoDR diabetic patients without DR.

However, there are some areas of incorrect segmentation as shown in Fig. 3 by yellow rectangles. AVA-Net performance can be improved by using larger datasets collected from patients with different diseases using different OCTA devices in different fields of view.

The AVA maps generated by AVA-Net are used to define three quantitative features, AA, VA, and AVAR, which are mathematically related. These quantitative features were calculated for the healthy control, NoDR, and mild DR groups. Our results indicate that quantitative features are significantly different between the control and mild DR groups. The mean AA value of healthy eyes in the whole image is 52.6%. This is significantly increased in mild DR eyes at 3.6%. The mean AVAR value for healthy eyes increased from 1.12 to 1.20 in mild DR eyes.

For AV analysis of the vascular perfusion density in OCTA-AV maps, four different quantitative features are also derived, named T-PID, A-PID, V-PID, and AV-PIDR, which do not require any thresholding method for binarizing the OCTA-AV maps. According to Fig. 6, significant decreases in T-PID, A-PID, and AV-PIDR were observed in mild DR eyes when compared with healthy eyes. Compared to NoDR eyes, V-PID showed a significant decrease in mild DR eyes. Compared to healthy eyes, NoDR eyes showed no differences in T-PID as a quantitative feature without differentiation of the AV areas, which is associated with total vessels, but showed decreases in A-PID and increases in V-PID. As a result of these opposite AV changes, the AV-PIDR, which is the ratio of the A-PID to the V-PID, is a sensitive quantitative feature to distinguish healthy, NoDR, and mild DR eyes from each other. The effectiveness of differential AV analysis can be seen here. Our results show that AV-PIDR of the NoDR and mild DR groups decreased by 5.1 and 2.6%, respectively, compared to the control group with a mean AV-PIDR value of 0.944. We also observed a 2.6% increase from NoDR to mild DR groups in the whole image AV-PIDR. Bonferroni correction indicates significant changes in A-PID from control to mild DR and in AV-PIDR from control to mild DR and from NoDR to mild DR. The combination of A-PID and AV-PIDR can provide supplementary information to each other and demonstrate significant changes in all three pairs according to Bonferroni correction. Therefore, we anticipate that ensemble analysis of AVA and PID features will allow robust detection and classification of DR and other eye diseases. A limitation of this study is the limited dataset size, therefore for some subjects both eyes were included in the statistical analysis of quantitative features. There may be some correlation between right and left eyes due to genetics and environmental factors.

## Conclusions
A deep learning network AVA-Net has been developed for robust AVA segmentation in OCTA images. The OCTA-AV map, which preserves perfusion intensity information for improved AV analysis, could be readily generated by multiplying the OCTA images by the AVA maps. Three area features, i.e., AA, VA, and AVAR were derived from the AVA maps, while four PID features, i.e., T-PID, A-PID, V-PID, and AV-PIDR were derived from the OCTA-AV maps. The three area features can reveal significant differences between the control and mild DR but cannot separate NoDR from mild DR and control groups. The PID features, T-PID and A-PID can differentiate mild DR from control but cannot separate NoDR from control and mild DR groups. V-PID can differentiate mild DR from NoDR but cannot separate control from NoDR and mild DR groups. In contrast, the AV-PIDR can disclose significant differences among all three groups, i.e., control, NoDR, and mild DR. According to Bonferroni correction, the combination of A-PID and AV-PIDR can demonstrate significant differences among all three groups.

## Data availability
The source data, i.e., the numerical results underlying the graphs and charts presented in the main figures, is provided as Supplementary Data 1. Training and test dataset images may be obtained from the corresponding author upon reasonable request.

## Code availability
The deep learning architectures, AVA-Net, are publicly available on GitHub[32].

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

## Acknowledgements

This project was funded by National Eye Institute (P30 EY001792, R01 EY023522, R01 EY030101, R01EY029673, R01EY030842), Research to Prevent Blindness, and Richard and Loan Hill Endowment.

## Author contributions

M.A. contributed to data preparation, network design, model implementation, data processing, statistical analysis and manuscript preparation. D.L. contributed to data preparation, data processing, statistical analysis and manuscript modification. B.E. contributed to data preparation, and network design. A.K.D. contributed to data preparation, data processing, and statistical analysis. J.I.L. contributed to data preparation, ground truth approval and manuscript modification. X.Y. supervised the project and contributed to study design and manuscript preparation. All authors reviewed and approved the manuscript.

## Competing interests

The authors declare no competing interests.
