## [Peer Review File · Communications Medicine]

Reviewers' comments:

Reviewer #1 (Remarks to the Author):

Manuscript Number COMMSMED-22-0468-T

"AVA-Net: an open-source deep learning network for arterial-venous area segmentation in OCT angiography"

This cross-sectional study introduced novel OCTA biomarkers for the early detection of people with diabetes without retinopathy and mild retinopathy from normal controls.

Major comments

They showed that the proposed OCTA biomarker was able to provide good detection. However, they should compare their approach (artery vein differentiated) with the conventional method (non-differentiated) as OCTA is well known to be affected in people with diabetes without retinopathy (PMID: 32503234).

They need to provide more information about the participants' characteristics in Table format. Factors such as age, signal strength or quality index of scans, systemic hypertension status (PMID: 34859021), and systemic blood pressure levels can affect the results. Also, the training and testing dataset need to be clarified.

Minor comments

They need to perform Bonferroni correction to counteract the multiple comparisons problem.

Please show some images that differentiate the seven novel parameters in the early detection of diabetes.

Figure 3 shows moderate, and severe DR. Patients with such eyes were not recruited for the study and should not be shown.

Reviewer #2 (Remarks to the Author):

The authors propose an automated deep learning architecture (AVA-Net) for artery/vein (AV) analysis in OCTA images. The proposed classifier goes beyond the artery/vein classification — already explored in the literature, by identifying artery/vein perfusion intensity areas, AV maps. A total of 178 scans of 6 mm x 6 mm field of view from healthy volunteers and patients with diabetes (with different severity of retinopathy) were included in the study. Ground truth artery/vein were manually identified using a set of rules previously established in another work (Akihiro Ishibazawa et al (2019)). Then, AV maps were generated using k-nearest neighbor (kNN) classifier. AVA-Net is a modified U-Net architecture for binary classification. Each pixel in the OCTA scan is either belonging to an arterial area or venous area. Using AVA maps obtained with the AVA-Net, the authors compute 7 retinal features

that might be informative for the discrimination of patient status. Hence, pairwise statistical tests are performed to investigate significant differences in those features across groups (controls, NoDR, mildDR).

The manuscript is well-written, it addresses an open problem in the retinal imaging field, and code is openly available. However, some clarifications are needed.

- 1) The generation of the ground truth (GT) can benefit from more detail. It seems that there are two sets of ground truth images obtained from two steps. The manually GT-AV images obtained using established rules to segment the large vessels, and then the GT-AVA maps generated using k-NN method applied to the GT-AV images. Who is classifying artery/vein in GT-AV images? Ophthalmologists, students, researchers?
- 2) The GT-AVA maps segmentation is obtained using kNN. Are those reviewed by an expert? How can we be sure that this segmentation is accurate?
- 3) Details about the dataset and its split in training and test are missing. Why do not include both eyes? Which eye (left or right) is chosen? Why NoDR participants are not included in the training (since this is a group of interest later on)? Why severe DR patients are not included in the test?
- 4) It is not very clear how some types of augmentation, e.g.; shifting, might be suitable for this binary classification. Can you please clarify how the types of augmentation are chosen and how they might affect the performances.
- 5) In line 252, the authors claimed that 5-CV is used to validate different architectures. Can you please clarify what you mean with different architectures?
- 6) In Figure 3 a severe DR case is shown. This patient does not have observable damage in the vasculature. It would be nice to show the AVA-maps on an OCTA scan with observable damage, e.g., a very sparse vascular network.
- 7) A circle of 1mm diameter center in the FAZ is removed from the final AV maps. How do you find the center of the FAZ?
- 8) T-test is appropriate for normally distributed variables. Is this the case?
- 9) As stated in the introduction, reference 1 suggests that vessel calibre is an important feature in diabetes. Can you please comment on why you are not investigating AV calibre in addition to the 7 features?
- 10) Figure 4 does not show the ground truth, why?
- 11) The claim in the line 80-81 "Using this approach, we can segment all the visible vessels in the OCTA images at different orders and scales as arterial or venous" seems an overstatement since the smallest capillaries are not really segmented. Can you please clarify that?
- 12) AV manual classification is obtained using rules valid for the 6mm x 6mm OCTA scans. Can the authors comment on the possibility to extend this investigation to 3mm x 3mm scans?

Subject: Response to review comments.

Manuscript Number: COMMSMED-22-0468-T

Manuscript title: AVA-Net: an open-source deep learning network for arterial-venous area segmentation in OCT angiography

Authors: Mansour Abtahi, David Le, Behrouz Ebrahimi, Albert K. Dadzie, Jennifer I. Lim, and Xincheng Yao.

We sincerely thank the editors, EBM, and reviewers for their valuable comments and constructive suggestions on the manuscript. In this revision, we have addressed all the review comments. Following is a summary of our response to the reviewers' questions/concerns. For your convenience, we cite each of the review comments before our response initiated with "Response".

Response to reviewer 1

Comment 1: They showed that the proposed OCTA biomarker was able to provide good detection. However, they should compare their approach (artery vein differentiated) with the conventional method (non-differentiated) as OCTA is well known to be affected in people with diabetes without retinopathy (PMID: 32503234).

Response: We have compared the total PID (T-PID) (conventional method without differential AV analysis) and the A-PID and V-PID (with differential AV analysis). In this revision, we made following clarifications:

Lines 190-192: "T-PID represents quantitative feature calculation without differentiation of the AV areas, while A-PID and V-PID represent quantitative features after AV area segmentation."

Lines 366-369: "By using the OCTA-AV map, all OCTA-related features previously reported in the literature such as VAD, vessel length density (VLD), vessel diameter index (VDI), vessel tortuosity (VT), and branchpoint density (BD) can be calculated separately for arterial and venous areas to check the effectiveness of the AV analysis."

Lines 394-399: "Compared to healthy eyes, NoDR eyes showed no differences in T-PID as a quantitative feature without differentiation of the AV areas, which is associated with total vessels, but showed decreases in A-PID and increases in V-PID. As a result of these opposite AV changes, the AV-PIDR, which is the ratio of the A-PID to the V-PID, is a sensitive quantitative feature to distinguish healthy, NoDR, and mild DR eyes from each other. The effectiveness of differential AV analysis can be seen here."

Comment 2: They need to provide more information about the participants' characteristics in Table format. Factors such as age, signal strength or quality index of scans, systemic hypertension status (PMID: 34859021), and systemic blood pressure levels can affect the results. Also, the training and testing dataset need to be clarified.

Response: Since this is a retrospective study, we do not have access to systemic blood pressure levels at the time of imaging, but we have other information that is added to the revised manuscript (line 113) 'Table 1. Demographic of the healthy subjects, NoDR and mild NPDR patients.' Our review of the quality index of the scans led us to exclude some scans that did not have a proper quality index. As a

result, the number of scans in the test dataset changed slightly. In the revision, we made following clarifications:

Lines 92-99: “In this study, we have two different datasets. The training dataset which consists of 104 OCTA images and their ground truths (68 control, 12 mild DR, 11 moderate DR, and 13 severe DR scans) is planned to be used for training and validation of the CNN. The test dataset which consists of 64 OCTA images without ground truths (25 eyes from 17 control participants, 18 eyes from 13 NoDR patients, and 21 eyes from 18 mild DR patients) is planned to be used for qualitative testing of the CNN and quantitative analysis with the focus on early detection of DR. Table 1 summarizes all participant demographics and diabetes-related parameters in the test dataset.”

Lines 104-106: “Subjects with macular edema, proliferative DR, previous vitrectomy surgery, history of other ocular disorders other than cataracts or minor refractive error, and ungradable and low-quality OCT pictures were exclusion criteria.”

Lines 205-213: “For statistical analysis of quantitative features, we treated each eye as a single observation. We performed the Shapiro–Wilk test to check if quantitative features are normally distributed. We used χ^2 tests to assess the distribution of gender and hypertension among different groups. Age and quality index distributions were compared using ANOVA. One-versus-one comparison of quantitative features between different groups was performed by the unpaired two-sided Student’s t-test. We also applied the Bonferroni correction to compare the difference in mean values of the quantitative features. $P < 0.05$ was considered statistically significant.”

Lines 305-310: “Studying the test dataset with demographic details presented in Table 1, no statistically significant differences were found between the different groups with regards to age, quality index, and gender (ANOVA, $P = 0.077$, ANOVA, $P = 0.079$, and χ^2 test, $P = 0.305$). Furthermore, there was no difference in hypertension across diabetes groups (χ^2 test, $P = 0.92$). Between the diabetic groups, there was a significant difference in the duration of diabetes (Student’s t-test, $P = 0.019$). The diabetic patients in this study were not insulin dependent.”

Comment 3: They need to perform Bonferroni correction to counteract the multiple comparisons problem.

Response: In this revision, we included the suggested Bonferroni correction (lines 316-320): “To reduce the occurrence of false positives in multiple hypothesis testing, we applied a Bonferroni correction as a conservative method for probability thresholding. Applying the Bonferroni correction, the statistical significance of the P value between the three groups was adjusted as $P < 0.0167$, and significant differences are denoted by the † symbol.” Following clarifications were added in the revision.

Lines 327-328: “The Bonferroni correction indicates that none of these above-mentioned changes are significant.”

Lines 341-344: “Based on the Bonferroni correction, changes in A-PID from control to mild DR and changes in AV-PIDR from control to mild DR and from NoDR to mild DR are significant. According to Bonferroni correction, A-PID and AV-PIDR together can show significant changes in all three pairs.”

Lines 402-406: “Bonferroni correction indicates significant changes in A-PID from control to mild DR and in AV-PIDR from control to mild DR and from NoDR to mild DR. The combination of A-PID and AV-PIDR can provide supplementary information to each other and demonstrate significant changes in all three pairs according to Bonferroni correction.”

Lines 419-420: “According to Bonferroni correction, the combination of A-PID and AV-PIDR can demonstrate significant changes in all three groups.”

Comment 4: Please show some images that differentiate the seven novel parameters in the early detection of diabetes.

Response: In this revision, we extended Fig. 1, by adding Fig. 1I-Fig. 1N, to illustrate the procedures of feature extraction.

Lines 199-200: “The processes for calculating these quantitative features are shown in Fig. 1 (I) to (N), with blue arrows and boxes.”

Comment 5: Figure 3 shows moderate, and severe DR. Patients with such eyes were not recruited for the study and should not be shown.

Response: As it is stated in the first paragraph of section 2.1 “Control subjects and diabetic patients with and without DR were recruited from the UIC retina clinic.”. To make it more clear, we changed this sentence as follows (lines 99-100): “Control subjects and diabetic patients without and with DR in different stages were recruited from the UIC retina clinic.”

Response to reviewer 2

Comment 1: The generation of the ground truth (GT) can benefit from more detail. It seems that there are two sets of ground truth images obtained from two steps. The manually GT-AV images obtained using established rules to segment the large vessels, and then the GT-AVA maps generated using k-NN method applied to the GT-AV images. Who is classifying artery/vein in GT-AV images? Ophthalmologists, students, researchers?

Response: In this study, Mansour Abtahi (researcher) labeled GT-AV maps and David Le (PhD student) checked the ground truths, which were confirmed by Jennifer Lim (ophthalmologist). As discussed at the beginning of section 2.2, readers can rely on the mentioned characteristics in OCTA images to manually detect arteries and veins accurately in the 6 mm × 6 mm dataset. The details of generating GT-AV maps for 6 mm × 6 mm and 3 mm × 3 mm OCTA images are reported in our previous work [1] which is cited in the first line of section 2.2.

[1] Abtahi, M., Le, D., Lim, J. I. & Yao, X. MF-AV-Net: an open-source deep learning network with multimodal fusion options for artery-vein segmentation in OCT angiography. *Biomedical Optics Express* 13, 4870-4888 (2022).

Comment 2: The GT-AVA maps segmentation is obtained using kNN. Are those reviewed by an expert? How can we be sure that this segmentation is accurate?

Response: Segmentation in GT-AVA maps is based on the segmentation of the vessels in the GT-AV maps. In this revision, we clarified that (lines 129-131): “Since we segmented all the visible large vessels in AV

maps and used kNN only to classify background pixels as pixels in arterial or venous areas, the generated AVA maps using kNN are reliable and accurate.”

Visual checking of the generated GT-AVA maps by Mansour Abtahi (researcher), David Le (PhD student), and Jennifer Lim (ophthalmologist) approved GT-AVA maps are sufficiently accurate.

Comment 3: Details about the dataset and its split in training and test are missing. Why do not include both eyes? Which eye (left or right) is chosen? Why NoDR participants are not included in the training (since this is a group of interest later on)? Why severe DR patients are not included in the test?

Response: The test dataset does not have ground truth and NoDR patients are in the test dataset. Our quantitative analysis focuses on the early detection of DR by comparing the quantitative features between control, NoDR, and mild DR groups in the test dataset. Because of this and some statistical analysis concerns we did not mix the training dataset with the test dataset for quantitative analysis. Thus, we modified these sentences in the first paragraph of section 2.1 (lines 92-98): “In this study, we have two different datasets. The training dataset which consists of 104 OCTA images and their ground truths (68 control, 12 mild DR, 11 moderate DR, and 13 severe DR scans) is planned to be used for training and validation of the CNN. The test dataset which consists of 64 OCTA images without ground truths (25 eyes from 17 control participants, 18 eyes from 13 NoDR patients, and 21 eyes from 18 mild DR patients) is planned to be used for qualitative testing of the CNN and quantitative analysis with the focus on early detection of DR.”

Left or right eyes may not have similar stages of DR or one of them may have exclusion criteria such as macular edema, proliferative DR, previous vitrectomy surgery, history of other ocular disorders other than cataracts or minor refractive error, and ungradable and low-quality OCT pictures as mentioned in section 2.1.

We added this sentence to the first paragraph of section 2.1 (lines 106-107): “There is no preference between left or right eyes.”

Comment 4: It is not very clear how some types of augmentation, e.g.; shifting, might be suitable for this binary classification. Can you please clarify how the types of augmentation are chosen and how they might affect the performances.

Response: In this revision, we clarified that (lines 262-266): “As retinal vessels in OCTA images have diverse tree-like patterns and differing vessel diameters, and because images can be taken from different locations of right and left eyes with diverse quality, the above-mentioned data augmentation methods could help improve the generalization capability of the CNN for segmenting unseen images.”

Comment 5: In line 252, the authors claimed that 5-CV is used to validate different architectures. Can you please clarify what you mean with different architectures?

Response: My apologies, this is a typo. We corrected in the revision (lines 281-282): “We performed 5-fold cross-validation on the training dataset, using 80 percent of the images for training and 20 percent of them for validating AVA-Net.”

Comment 6: In Figure 3 a severe DR case is shown. This patient does not have observable damage in the vasculature. It would be nice to show the AVA-maps on an OCTA scan with observable damage, e.g., a very sparse vascular network.

Response: Thank you for your suggestion. However, another reviewer suggested us to remove moderate and severe images from the manuscript, because the current study is primarily focusing on the demonstration of using AV analysis for objective differentiation of healthy, NoDR, and mild DR images.

Comment 7: A circle of 1mm diameter center in the FAZ is removed from the final AV maps. How do you find the center of the FAZ?

Response: In this revision, we clarified that (lines 157-159): “During image scanning of the macula, the commercial imaging device detects the fovea center automatically to keep the fovea at the center of the image. Accordingly, we can say that the center of the circle is approximately in the center of the image.”

Comment 8: T-test is appropriate for normally distributed variables. Is this the case?

Response: In this revision, we clarified that (lines 206-207): “We performed the Shapiro–Wilk test to check if quantitative features are normally distributed.” Also, we added the following sentence to the second paragraph of section 3.2 (lines 313-314): “Shapiro–Wilk test indicated that all features were normally distributed.”

Comment 9: As stated in the introduction, reference 1 suggests that vessel calibre is an important feature in diabetes. Can you please comment on why you are not investigating AV calibre in addition to the 7 features?

Response: In this study we tried to introduce 7 new quantitative features that are related to AVA maps and OCTA-AV maps. AV analysis of previously reported quantitative features such as vessel diameter index (VDI) can be one of our future studies. To better clarification we modified a sentence in the second paragraph of section 4 (lines 366-369) as follows: “By using the OCTA-AV map, all OCTA-related features previously reported in the literature such as VAD, vessel length density (VLD), vessel diameter index (VDI), vessel tortuosity (VT), and branchpoint density (BD) can be calculated separately for arterial and venous areas to check the effectiveness of the AV analysis.”

Comment 10: Figure 4 does not show the ground truth, why?

Response: In this revision, we clarified that (lines 94-98): “The test dataset which consists of 64 OCTA images without ground truths (25 eyes from 17 control participants, 18 eyes from 13 NoDR patients, and 21 eyes from 18 mild DR patients) is planned to be used for qualitative testing of the CNN and quantitative analysis with the focus on early detection of DR.”

We also modified the last sentence in section 3.1 (lines 291-293) as follows: “The visual performance of AVA-Net for segmenting representative OCTA images in the test dataset containing only OCTA images is presented in Fig. 4.”

Comment 11: The claim in the lines 80-81 “Using this approach, we can segment all the visible vessels in the OCTA images at different orders and scales as arterial or venous” seems an overstatement since the smallest capillaries are not really segmented. Can you please clarify that?

Response: That seems true. Technically by generating OCTA-AV maps, we are not segmenting every capillary, but we are classifying every visible capillary in the OCTA images as an artery or vein which are included in the arterial or venous regions, respectively. We changed “segment” to “classify” in this sentence (lines 80-81) as follows: “Using this approach, we can classify all the visible vessels in the OCTA images at different orders and scales as arterial or venous.”

Comment 12: AV manual classification is obtained using rules valid for the 6mm x 6mm OCTA scans. Can the authors comment on the possibility to extend this investigation to 3mm x 3mm scans?

Response: Generating GT-AV maps for 3mm x 3mm scans is discussed in our previous work [1] which is cited in the first line of section 2.2. To address this comment, we added these sentences to the end of section 2.2 (lines 138-140) in the revised manuscript as follows: “Generating ground truth AV maps for 3 mm × 3 mm OCTA images is discussed in our previous study²⁴. The above-mentioned procedure can be used to generate ground truth AVA maps for 3 mm × 3 mm OCTA images.”

REVIEWERS' COMMENTS:

Reviewer #1 (Remarks to the Author):

I am satisfied with the revision and do not have further comments.

Reviewer #2 (Remarks to the Author):

The authors have addressed all comments/ concerns previously raised. However, before publication I would suggest to revise the following points:

1. Reference 1 cites "An Automatic Graph-Based Approach for Artery/Vein Classification in Retinal Images" a manuscript that describes a method for A/V identification in fundus images. Whereas the place of the citation in the text seems to refer to a study that report a change in vessel diameter in diabetes. Please be sure that reference 1 is correct and in the right position in the text.
2. In lines 131-133, I suggest to add that the final maps were reviewed and approved by an ophthalmologist.
3. Please increase size of Figure 1.
4. In lines 194-195 "T-DIP represents quantitative ..." was repeated twice.
5. In line 208, the authors assert that each eye is treated as a single observation. However, genetic and environmental factors affect similarly both eyes, and ocular intracorrelation should be taken into account in statistical analysis. Please justify your choice in paragraph 2.4 and state this as a limitation of you study in the discussion.
6. After the further clarifications about the training and test datasets there is no need to modify the last sentence of paragraph 3.1.
7. Line 312 "The diabetic patients in this study were not insulin dependent" goes into Data acquisition.

Subject: Response to review comments.

Manuscript Number: COMMSMED-22-0468A

Manuscript title: AVA-Net: an open-source deep learning network for arterial-venous area segmentation in OCT angiography

Authors: Mansour Abtahi, David Le, Behrouz Ebrahimi, Albert K. Dadzie, Jennifer I. Lim, and Xincheng Yao.

We sincerely thank the editors, EBM, and reviewers for their valuable comments and constructive suggestions on the manuscript. In this revision, we have addressed all the review comments. Following is a summary of our response to the reviewers' questions/concerns. For your convenience, we cite each of the review comments before our response initiated with "Response".

Response to reviewer 2:

The authors have addressed all comments/ concerns previously raised. However, before publication I would suggest to revise the following points:

Comment 1: Reference 1 cites "An Automatic Graph-Based Approach for Artery/Vein Classification in Retinal Images" a manuscript that describes a method for A/V identification in fundus images. Whereas the place of the citation in the text seems to refer to a study that report a change in vessel diameter in diabetes. Please be sure that reference 1 is correct and in the right position in the text.

Response: In the introduction of the cited paper, vessel diameter alterations in the early stages of diabetic retinopathy is mentioned. In the revision, we added another citation with quantitative analysis on the vessel diameter alterations in different stages of diabetic retinopathy.

Comment 2: In lines 131-133, I suggest to add that the final maps were reviewed and approved by an ophthalmologist.

Response: In the revision, we added the suggested clarification (lines 146-147): "They were reviewed and approved by an ophthalmologist."

Comment 3: Please increase size of Figure 1.

Response: In the revision, we enlarged the size of Figure 1.

Comment 4: In lines 194-195 "T-DIP represents quantitative ..." was repeated twice.

Response: Two sentences are slightly different. In the revision, we combined them to one sentence (Lines 196-198): "T-PID represents quantitative feature calculation without differentiation of the AV areas,

while A-PID and V-PID represent quantitative features after AV area segmentation for differential AV analysis.”

Comment 5: In line 208, the authors assert that each eye is treated as a single observation. However, genetic and environmental factors affect similarly both eyes, and ocular intracorrelation should be taken into account in statistical analysis. Please justify your choice in paragraph 2.4 and state this as a limitation of your study in the discussion.

Response: In the revision, we made following modifications:

Section 2.4 (Lines 210-211): “For the statistical analysis of quantitative features, due to the limited dataset size, we treated each eye as a single observation for some subjects with images of both eyes.”

Section 4 (Lines 388-391): “A limitation of this study is the limited dataset size, therefore for some subjects both eyes were included in the statistical analysis of quantitative features. There may be some correlation between right and left eyes due to genetics and environmental factors.”

Comment 6: After the further clarifications about the training and test datasets there is no need to modify the last sentence of paragraph 3.1.

Response: In this revision, we returned this sentence to the original version (lines 289-291): “The visual performance of AVA-Net for segmenting representative OCTA images in the test dataset is presented in Fig. 4.”

Comment 7: Line 312 “The diabetic patients in this study were not insulin dependent” goes into Data acquisition.

Response: In this revision, we added this sentence to section 2.1 (lines 120): “The diabetic patients were not insulin dependent.”